# Self-organization of an inhomogeneous memristive hardware for sequence learning

Melika Payvand [1,4] ✉, Filippo Moro[1,2,4], Kumiko Nomura [3], Thomas Dalgaty [2], Elisa Vianello [2], Yoshifumi Nishi[3] & Giacomo Indiveri [1]

Learning is a fundamental component of creating intelligent machines. Biological intelligence orchestrates synaptic and neuronal learning at multiple time scales to *self-organize* populations of neurons for solving complex tasks. Inspired by this, we design and experimentally demonstrate an adaptive hardware architecture *Memristive Self-organizing Spiking Recurrent Neural Network (MEMSORN)*. MEMSORN incorporates resistive memory (RRAM) in its synapses and neurons which configure their state based on Hebbian and Homeostatic plasticity respectively. For the first time, we derive these plasticity rules directly from the statistical measurements of our fabricated RRAM-based neurons and synapses. These "technologically plausible" learning rules exploit the intrinsic variability of the devices and improve the accuracy of the network on a sequence learning task by 30%. Finally, we compare the performance of MEMSORN to a fully-randomly-set-up spiking recurrent network on the same task, showing that self-organization improves the accuracy by more than 15%. This work demonstrates the importance of the device-circuit-algorithm co-design approach for implementing brain-inspired computing hardware.

The hallmark of intelligence is the ability of the brain to adapt and self-organize itself to sensory information it receives throughout its lifespan. This self-organization is mediated by a rich set of neuro-cognitive mechanisms that together contribute to sequence learning and long-term memory formation[1]. While learning, a web of memory forms between large groups of neurons, leading to coherent dynamic activity patterns that are a function of the sensory information the system receives.

It has been shown that the combination of brain-inspired learning rules at different time scales lends themselves to the self-organization of dynamic networks for behavior control[2,3]. This type of self-organization lies in the unsupervised learning realm where the ground truth is not available for learning. Instead, the memory forms as a result of clustering information in cell-assemblies[2]. A cell assembly can be defined as a group of neurons with strong mutual excitatory connections. Once a subset of a cell assembly is stimulated, its neurons

tend to be activated as a whole, so that the cell can be considered as an operational unit of a Spiking Recurrent Neural Network (SRNN). Applying local learning rules to the recurrent connections forms independent cell assemblies and makes the SRNN more powerful in extracting temporal features in the data, compared to a fully-randomly-connected solution[3].

One example of such a concept has been shown in Self-Organizing Recurrent Network (SORN)[3], a recurrent network model of excitatory and inhibitory binary neurons. It incorporates Hebbian-based synaptic plasticity at a shorter timescale, along with Homeostatic plasticity at a longer timescale. It is illustrated that SORN outperforms random Recurrent Neural Networks (RNNs) without plasticity on sequence prediction tasks.

Implementing SORN-like networks on a hardware substrate holds great promise for machine intelligence and autonomous agents, especially in situations where the agent is in unknown environments[4,5].

[1]Institute for Neuroinformatics, University of Zurich and ETH Zurich, Zurich, Switzerland. [2]Université Grenoble Alpes, CEA, Leti, F-38000 Grenoble, France. [3]Corporate Research & Development Center, Toshiba Corporation, Kawasaki, Japan. [4]These authors contributed equally: Melika Payvand, Filippo Moro. ✉e-mail: melika@ini.uzh.ch

Neuromorphic technologies with online learning capabilities can support the hardware implementation of such self-organizing SRNNs[6,7].

Online learning in electronic devices requires local and distributed memory elements for storing the learned parameters (e.g., the synaptic weights). Resistive Random Access Memory (RRAM) has recently gained significant attention as a promising memory technology for on-line learning[8–16]. Its non-volatile and multi-state properties make it a plausible candidate for employment in adaptive hardware. Importantly, its internal dynamics and intrinsic stochasticity have been proven beneficial for on-chip learning[17–20] which cannot be simply introduced in a digital implementation[6,21]. As biological networks rely on small unreliable components for reliable learning, they can provide guidance for learning with RRAM devices. Brain-inspired unsupervised Hebbian learning strategies have already been extensively explored in adaptive memristive neuromorphic architectures[12,22–24]. In these works, the RRAM conductance changes towards a more/less conductive state based on the correlation/anti-correlation between its pre- and post-synaptic neurons. However, Hebbian learning by itself cannot robustly lead to self-organization, as it implements a greedy mechanism which can lead to unstable dynamics[25]. To achieve self-organization in memristive neuromorphic architectures, a multitude of plasticity mechanisms need to be at play together, with properties and dynamics that match the physics of the underlying adaptive hardware substrate[26,27].

Here we present *MEMSORN*: a hardware architecture inspired by SORN with multi-timescale on-chip plasticity rules. MEMSORN is developed following a device-algorithm co-design approach exploiting the physics of the employed RRAM devices taking advantage of their variability.

We designed and fabricated the RRAM-based synapse and neurons in 130 nm CMOS technology integrated with $HfO_2$-based RRAM devices. Based on the statistical measurements from these designs, we derive the local technologically plausible plasticity mechanisms (Hebbian and Homeostatic), and apply them in the MEMSORN architecture. We benchmark the network on a sequence learning task and show that this approach exploits the intrinsic variability of the RRAM devices and improves the network's accuracy as a function of sequence length, learning rate, and training epochs. As a control experiment, we apply the same task to the same exact network, only without learning, whose recurrent connections are randomly set up. We show that MEMSORN accuracy outperforms the random network by about 15%. This work represents a fundamental step toward the design of future neuromorphic intelligence devices and applications.

## Results

Inspired by SORN[3], we implemented two recurrently-connected networks of Leaky Integrate and Fire (LIF) neurons: one randomly connected with fixed weights (static) and one with connections that change through learning (MEMSORN). Other than this difference, the two networks are identical. Both networks consist of an excitatory pool of recurrently connected neurons, and an inhibitory pool of neurons that inhibit the excitatory ones, along with a read-out layer fully connected to the two pools. The inhibitory neurons balance the activity of excitatory neurons by providing a negative feedback[28,29]. Inspired by neuro-anatomy considerations on cortical circuits, we divided the excitatory and inhibitory population into 80% and 20% of the total number of neurons, respectively[30] (see Methods). Different sub-populations of neurons are stimulated by different parts of the input sequence. In both networks, the activities of all the recurrent neurons are fed to a linear classifier at the readout which learns to distinguish between different classes of input (see Fig. 1a).

### Hardware Implementation

**Architecture.** To implement the network in hardware, we designed a crossbar memory architecture (Fig. 1b). Its rows are connected to the neurons and its columns are connected to either external inputs or to a recurrent input from another neuron. We employed RRAMs both in the design of the synapses at the cross-points holding their strength (Fig. 1b), and in the design of the neurons holding their internal

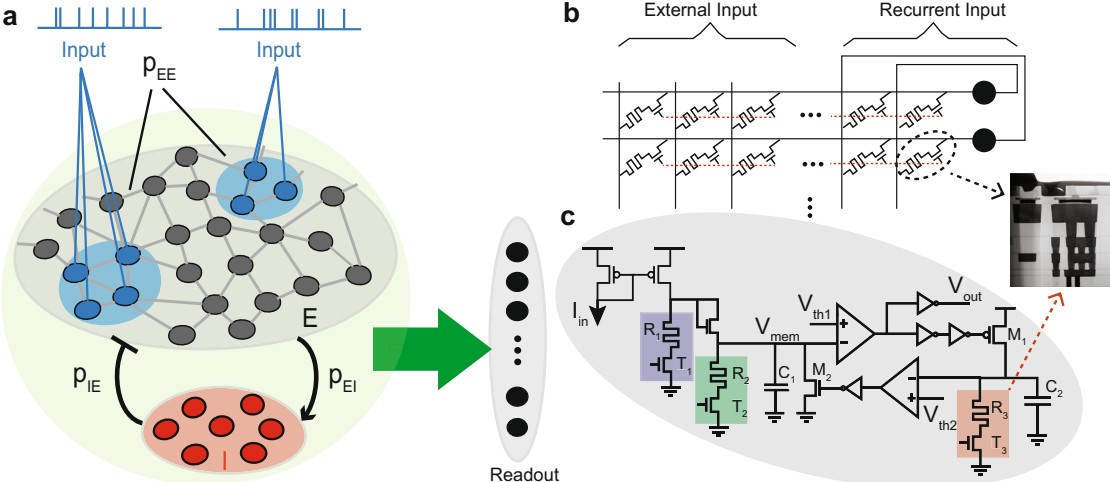

**Fig. 1 | Spiking Recurrent Neural Network (SRNN) and its hardware implementation. a** The SRNN is composed of a recurrent pool of excitatory neurons (E) whose connections are formed by random fixed weights (static) or through learning (MEMSORN). The probability of these connections are defined as $p_{EE}$. The network is excited by spatio-temporal inputs activating sub-populations, shown in blue. Each of the sub-populations encodes a particular part of the sequence. The excitatory (E, gray) and inhibitory (I, red) populations are connected to each other with probabilities $p_{IE}$ and $p_{EI}$. There are no recurrent connections amount the inhibitory populations. Both populations contribute to the activation of the output, via the readout connections (green arrow). Each neuron in the readout is assigned to a different prediction class. **b** A possible hardware implementation of the SRNN.

Neuron's recurrent and external input connections are implemented by RRAM devices assembled in a crossbar array. Rows of the crossbar are connected to the inputs, while its columns are connected to the neurons. **c** Neurons are implemented using a hybrid CMOS/RRAM design. RRAMs hold the parameters of the neurons, such as gain ($T_1 - R_1$, purple), leak ($T_2 - R_2$, green), and refractory period ($T_3 - R_3$, red). Neuron integrates part of $I_{in}$, defined by the gain, on its membrane capacitance, $C_1$, giving rise to a potential $V_{mem}$. As soon as $V_{mem}$ passes the neuron threshold, $V_{th1}$, it generates a pulse at $V_{out}$. The output pulse charges $C_2$, whose voltage rises above $V_{th2}$, putting the neuron in the refractory regime, closing transistor $M_2$, and resetting $C_1$.

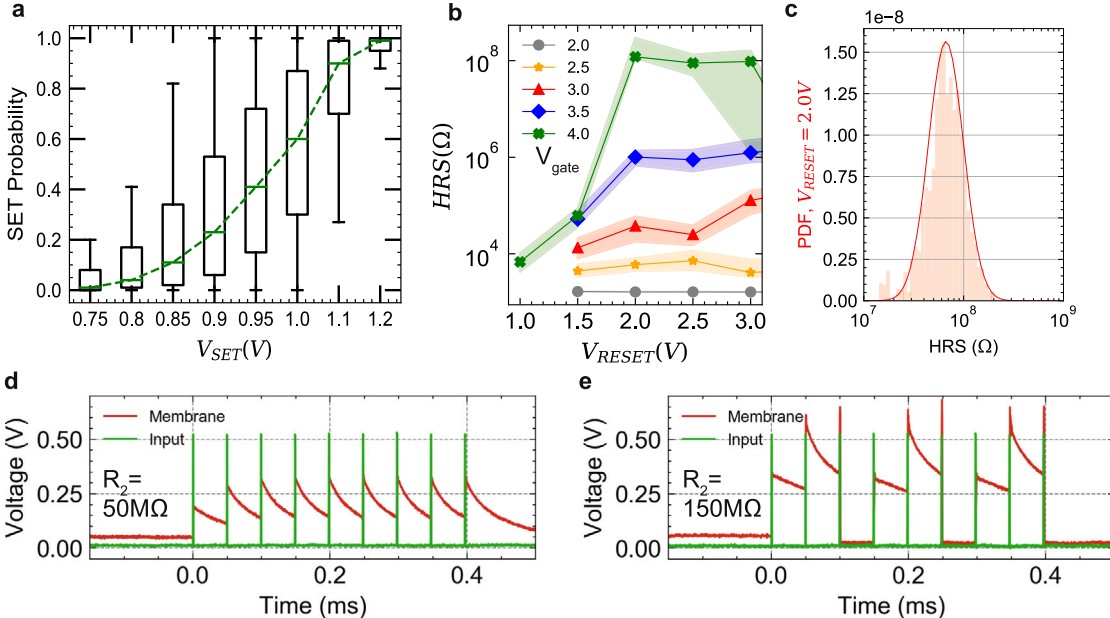

**Fig. 2 | Measurements from our fabricated synapse and neurons of Fig. 1 in 130 nm technology integrated with HfO₂-based RRAM. a–c** Experimental results from the fabricated 4 kb synapse array, each device is programmed 100 times. **a** SET characteristics; The box plot represents the SET probability as a function of the SET voltage, over the RRAM population; the green horizontal bar represents the median value, the box lower and upper limits represent the ± 25% and ± 75% quartile respectively, and the whiskers show the ± 95% quartile. The dashed green line connecting the median values shows the emerging sigmoidal behavior of the SET probability over the SET voltage. **b** RESET characteristics; HRS measurements as a function of the RESET voltage applied across the devices ($V_{RESET}$), for different gate

voltages applied to the transistor (T) ($V_{gate}$). **c** The HRS distribution at $V_{RESET}$ = 2.0 $V$ and $V_{gate}$ = 4 $V$ which fits well to a log-normal distribution, and we used in our neuron model. **d**, **e** Experimental results from the fabricated neuron. The neuron is excited by a train of spikes with a pulse width of 1 $\mu s$ and a magnitude of 450 $mV$ and a frequency of 1 $kHz$ (green). Neuron membrane potential is measured (red). Changing the state of $R_1$ and $R_2$ devices changes the gain and time constant of the neuron. **d**, **e** Neuron's time constant and thus its firing rate changes by changing R2. Gain is set so as to make the neuron integrate for many pulses before it fires. In (**d**), $R_2$ is set to 50 $M\Omega$, and in (**e**) it is set to 150 $M\Omega$, increasing the time constant, making the neuron fire more often.

parameters (Fig. 1c); Each synapse contains a transistor in series with an RRAM (aka 1T-1R), with the free side of the transistor and the RRAM connecting to the rows and the columns, respectively; Each neuron implements the LIF model shown in Fig. 1c. This hybrid CMOS/RRAM neuron design encompasses three RRAMs whose value set the neuron time constant (shown in green), gain (shown in purple), and refractory period (shown in red) (see Methods)[31]. The adaptive nature of RRAM allows for learning both the synaptic and internal neuron parameters in an on-chip and online fashion.

As soon as an input spike arrives at a column, a voltage is applied across the corresponding synaptic RRAMs, giving rise to a current, through Ohm's law. All currents are summed at the rows and are integrated by the corresponding neurons[18]. The input to the neuron is multiplied by the gain ($R1/R2$) and is integrated on the membrane capacitance $C_1$ with a time constant determined by $R_2C_1$. As soon as the voltage on $C_1$ passes the threshold $V_{th1}$, the neuron generates a voltage spike, and sends it both to $V_{out}$ and to the feedback path. In the feedback path, the neuron's spike is integrated on $C_2$, and the resulting voltage has a time constant of $R_3C_2$. As soon as this voltage passes the threshold $V_{th2}$, the membrane capacitance $C_2$ is reset, and the neuron awaits the next input current.

**Synapse and neuron characteristics.** We fabricated and measured a 4 kb synaptic crossbar array along with the hybrid CMOS/RRAM neurons, using 130 nm CMOS technology integrated with HfO₂-based RRAMs.

In the synapses, we can induce a change by applying a voltage across the RRAM devices. The device state changes from a High-Resistive State (HRS) to a Low-Resistive State (LRS) (SET operation) by applying a positive voltage between the positive and negative terminals of the 1T-1R, while applying a voltage to the gate of the transistor, $V_{gate}$, to control the current passing through it during programming.

Alternatively, the device switches from LRS to HRS, by applying a negative voltage across the 1T-1R (RESET operation). Both SET and RESET operations produce changes in a stochastic manner. This results in a distribution over the resistance values given a programming condition[31–33]. We define a threshold at 50 $k\Omega$ for the resistance marking the border between HRS and LRS, and characterize the SET and RESET properties; Fig. 2a shows the probability of the SET operation as a function of the voltage applied to the 1T-1R cell, which follows a sigmoidal function. The RESET operation is characterized in Fig. 2b as a function of the voltage applied across the devices, with different gate voltages. The distribution of HRS values for a RESET voltages of 2 $V$ is shown in Fig. 2c. The distribution fits well with a log-normal function[33].

In the neurons, we measured the output firing pattern in response to a spike train as is shown in Fig. 2d and e. Setting $R_2$ with lower values increases (Fig. 2d), and with higher values decreases (Fig. 2e, f) the neuron's time constant, and thus changes the likelihood of the neuron firing. In sensory-motor applications, matching the dynamics of sensory signals to those of the electronic circuits in the processing hardware can minimize the system power consumption and maximize the Signal to Noise Ratio (SNR)[4]. Therefore, to obtain neuron's time constants of millisecond range, on the order of sensory signals, while limiting the size of the capacitors (to minimize area usage), the neuron's RRAM devices should be operated in their HRS ranging from M Ω to G Ω (Fig. 1b).

**Technologically-plausible algorithms**

With the technologically plausible algorithm design, we aim to optimize the hardware implementation of algorithms by taking the hardware physics into account while developing the algorithm. Figure 3 depicts the algorithms for the two static and MEMSORN networks which are derived based on the synapse and neuron measurements of Fig. 2.

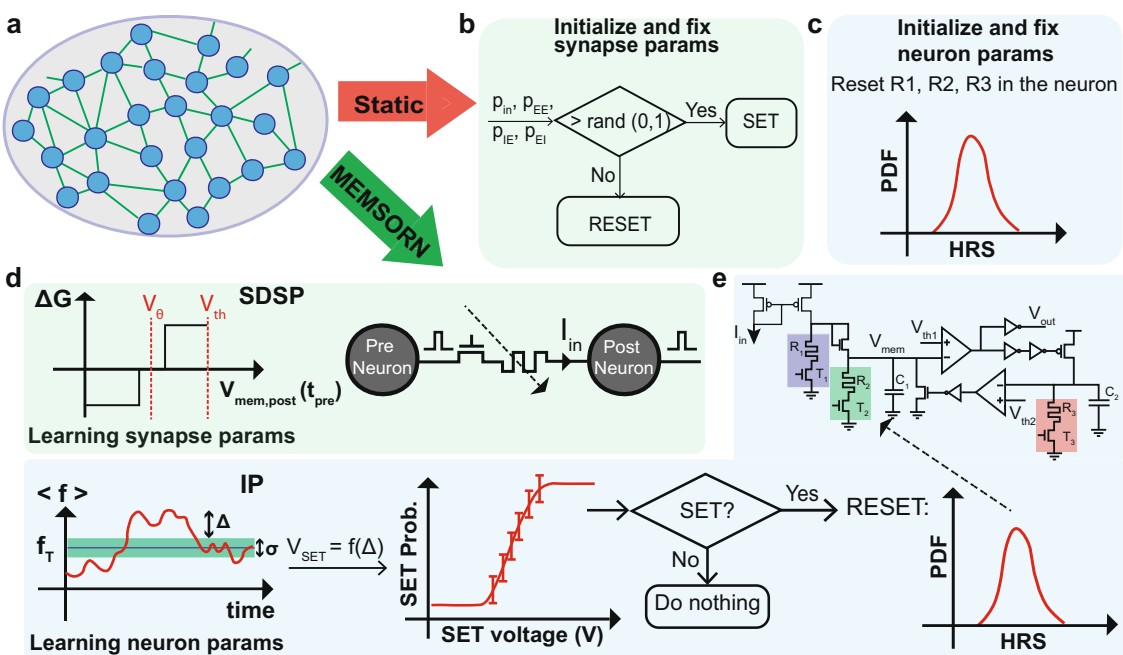

**Fig. 3 | Technologically plausible algorithms for static and MEMSORN networks.** **a** A Spiking Recurrent Neural Network (SRNN) with excitatory connections. **b, c** Synapse and neuron parameters for the static network. Both values are fixed after an initialization process. **b** Synaptic parameters are initialized based on comparing the probability of different connections with a random number. **c** Neuronal parameters are initialized by resetting the memristors $R_1$, $R_2$ and $R_3$ which is equivalent to sampling from a log-normal distribution around a mean resistance that is a function of the reset voltage. **d, e** Synaptic and neuron parameters for MEMSORN network. Both parameters are learned throughout the input sequence presentation. **d** Synaptic parameters are learned based on the Hebbian-based Spike Dependent Synaptic Plasticity (SDSP) learning rule. At the time of the pre-synaptic event ($t_{pre}$), weight (conductance) of the synapses (RRAMs) are increased/decreased, if the membrane potential of the post-synaptic neuron ($V_{mem,post}$) is higher/lower than $V_\theta$. **e** Neuron parameters are changed based on the IP algorithm which tries to keep the firing rate of each neuron ($<f>$) in a healthy regime ($f_T \pm \sigma/2$). If the neuron's firing rate goes beyond this regime, neurons' $R_2$ is first SET probabilistically and then it is RESET. The RESET process samples a new value for $R_2$ from the log-normal distribution of HRS values.

**Static network.** The algorithm for the static network (i.e., with fixed random weights) is depicted in Fig. 3b, c. The synapse and neuron behavior is fixed a priori; the synaptic connections are set randomly, by comparing the probability of connections in different populations to a random number, and if higher/lower, induce a SET/RESET to the devices (Fig. 3b, See Methods for details); the neuron parameters are sampled from the HRS log-normal Probability Distribution Function (PDF) derived from the measurements of Fig. 2c, equivalent to applying $V_{RESET} = 2V$ to the devices.

**MEMSORN.** The MEMSORN plastic network self-organizes to form multiple cell-assemblies. This is done by changing the RRAMs in the synapse and neuron parameters through learning. The excitatory synapses undergo a Hebbian-type plasticity, i.e., Spike Driven Synaptic Plasticity (SDSP), which changes the synaptic RRAM based on the correlation between the input (pre-synaptic) and output (post-synaptic) neural activities[7,34]. In addition, the neuron parameters undergo Homeostatic plasticity, i.e., Intrinsic Plasticity (IP), which acts as a regulatory mechanism to keep the neuron's firing activity within the desired range[35]. Both forms of plasticity are well suited for the implementation on CMOS and RRAM hardware.

Following the SDSP rule, the RRAM resistance of a synapse is decreased/increased, on the onset of its pre-synaptic spike, if the membrane potential of the post-synaptic neuron is higher/lower than $V_\theta$ threshold (Fig. 3d).

On the other hand, IP changes the neuron's RRAM to maintain its output firing rate, $f_n$, close to a target firing frequency, $f_T$, within a tolerance of $\sigma$ (Fig. 3e). If $f_n$ lies outside of these boundaries, the RRAMs in their HRS are updated accordingly. For simplicity, we have chosen to only update $R_2$ which simultaneously changes both the gain and the

time constant of the neuron. Changing the gain will additionally implement synaptic scaling which is another homeostatic plasticity mechanism, used in conjunction with IP in the original SORN paper[3]. To tune $R_2$ in HRS, it is first SET and then RESET. SET is done probabilistically proportional to the difference between $f_n$ and $f_T$ ($\delta$). Once SET, The RESET operation with a fixed $V_{RESET}$ effectively samples a new HRS value from a log-normal PDF. Therefore, neurons with a frequency deviating significantly from the target will change their leak and gain by acting on $R_2$, to adapt their firing rate. Note that since the amplitude of $V_{RESET}$ is fixed, the sampled HRS value is drawn from a single distribution, which makes the search for the correct resistance values non-guided. This reduces circuitry overhead with respect to an alternative algorithm in which the RESET operation is performed by adapting the $V_{RESET}$ to the deviation of the $f_n$ from $f_T$ (i.e. $V_{RESET} \propto |f_n - f_T|$)[35] (see Methods).

## Benchmark

To validate our approach, we used the same benchmark proposed in the original SORN paper[3]: a sequence learning task based on counting for predicting the next sequence at the output. The network receives a shuffled alternation of two input sequences of length $n+2$ of 6 possible characters in $[A, B, C, D, E, F]$. In both sequences, either characters, $B$ or $E$ are repeated $n$ times (Fig. 4a). Examples of these sequences are $S_{1,n}$: $[A, B_1, B_2, \ldots, B_n, C]$ and $S_{2,n}$: $[D, E_1, E_2, \ldots, E_n, F]$. The goal is to learn to predict the next character given all the previously-presented ones, *i.e* $P(next-character_i | \sum_{j}^{i-1} shown-character_j)$. After fixing the length of the sequence, the network has to learn to count the repetition of characters $B$ and $E$ by means of a reliable dynamical state.

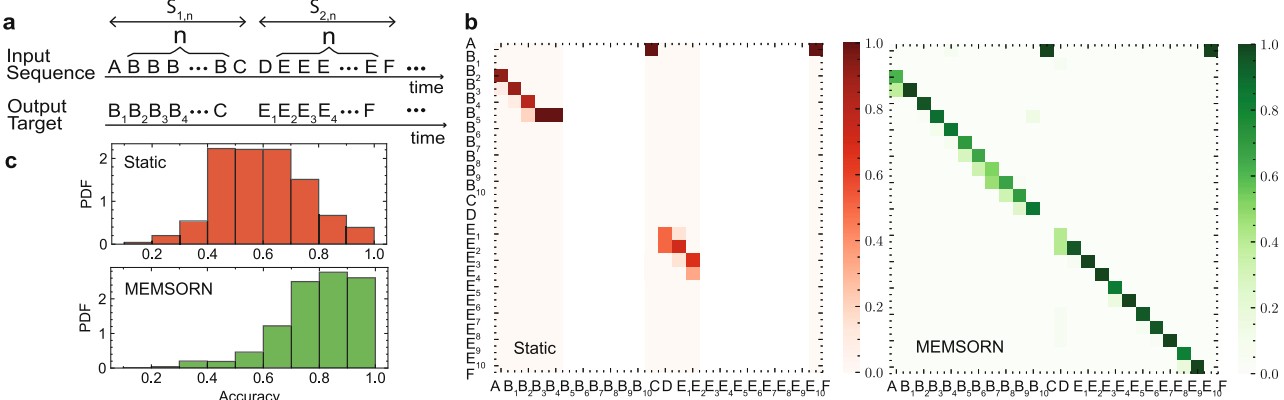

**Fig. 4 | Static and MEMSORN performance comparison. a** Sequence learning task. Two input sequences of $S_{1n} = ABB...BC$ and $S_{2n} = DEE...EF$, with $B$s and $E$s presented $n$ times, are fed to the network. Each letter represents part of the sequence. The task is to predict the next symbol in the sequence, for which it needs to keep the count of the presented $B$ and $E$ symbols. **b** Confusion matrix for the static (red) and MEMSORN (green) networks. The static network is capable of separating only the first part of the sequence, whereas the MEMSORN network can successfully predict the next letter in the sequence resulting in a diagonal confusion matrix. Understandably, the performance drops to chance level between the two sequences, since input symbols are equiprobable and their temporal succession carries no structure. **c** Histogram of the accuracy in the static (red) and MEMSORN (green) networks tested on 1000 different networks, for the counting task with sequence length ($n$) of 10. The mean of the accuracy distribution in MEMSORN network increases compared to the static network (mean of 0.756 for MEMSORN network compared to 0.596 in the static network). Also, the number of low-accuracy networks in MEMSORN is greatly reduced compared to the static network (about four times).

We applied the counting task to the static and MEMSORN networks and compared their performance. The network is asked to differentiate between $n = 10$ repetitions of the same symbol, presented in the middle of the two sequences. Each symbol's position in the two sequences is assigned to one output neuron in the readout whose activity represents the network's prediction of the next symbol. Fig. 4b shows the confusion matrix, indicating the predicted, compared to the expected output.

The static network is capable of separating only the first repetitions, whereas the MEMSORN network can successfully resolve all the repetitions, forming a diagonal in the confusion matrix, matching the output to the target. Since the two sequences are randomly alternated, the output of the network under the presentation of the last symbol in the sequence cannot be predicted. This applies to outputs 0, 13, and 25 in Fig. 4b. The internal dynamics in the static network saturate and lands on an attractor state from which no further information can be extracted. The MEMSORN network, instead, is capable of forming more complex dynamics that allow for fading memory to form and separate the repetitions in the input sequence. Figure 4c illustrates the histogram of the accuracy calculated over 1000 networks initialized differently for both networks on the counting task with the sequence length of $n = 10$. As shown, the mean accuracy of the MEMSORN network is increased compared to the static network. (Mean accuracy of 0.756 compared to 0.596 respectively). The standard deviation is due to the random initialization of the connections and the variability of RRAMs, implementing both the weights of the connections and the parameters of the neurons (See Methods.) Taking into account the hardware constraints, our statistical analysis shows that by enabling learning inside the recurrent network, there is a higher probability of obtaining a more accurate network; i.e. the number of learned networks that can correctly predict the next letter with an accuracy of more than 0.8, is four times that of the static network.

**Analysis on the effect of variability in MEMSORN**
RRAM devices undergo cycle-to-cycle and device-to-device variability as was confirmed with our measurements in Fig. 2. To understand the effect of variability in MEMSORN, we performed simulations on four cases: (i) No device variability and IP operation off; (ii) Variability in devices receiving the SDSP rule, and IP operation off; (iii) Variability in devices receiving the SDSP, and IP operation on without variability; (iv)

Variability in both SDSP and IP learning with standard deviation for the IP devices set as 0.1, taken from our measurements. It is worth noting that in condition (iii), since there is no variability in IP operation, the same initial value is always applied when IP is acting.

Figure 5 shows the network performance under these four conditions. The figure demonstrates the positive effect of the regularizing IP mechanism, and how MEMSORN network exploits the different sources of variability of the RRAM devices to increase its accuracy on the sequence learning task; Figure 5a plots the histograms of accuracy for every 100 samples of learning in the MEMSORN network for all the variability conditions. The histograms show that introducing IP operation, and any source of variability shifts the mean of the accuracy of the network to higher values; Figure 5b illustrates the accuracy as a function of the sequence length. As the sequence length increases, the network needs to remember increasingly longer sequences which tests its fading memory[36]. Thus, the accuracy of the network drops with longer sequences. It is worth noting that as the sequence length increases, the number of output neurons increases, and thus the baseline chance level accuracy reduces. Figure 5b confirms that the networks including IP and added source of variability outperform other conditions. Figure 5c depicts the network accuracy as a function of the SDSP learning rate (See Methods).

Despite that a large learning rate results in a consistent drop in accuracy, introducing variability suppresses accuracy degradation. This suggests that the noise introduced by the variability of the RRAM devices is beneficial for the stability of the network making it less sensitive to hyper-parameters and low bit resolution. This is because through learning with noise, the algorithm finds a set of parameters that are more insensitive to noise. Figure 5d shows the accuracy evolution of the MEMSORN network during learning epochs. Each epoch consists of presenting one of the two sequences which are presented to the network with a random order. Condition (i) without any variability and IP operation (black) leads to more stable learning dynamics, but also lower performance. Instead, adding noise to SDSP or adding the IP operation causes some instability in the network, but also allows for much higher overall accuracy. Finally, combining the variability in SDSP with that of IP leads to the best performance compared to other conditions.

The positive effect of variability is because a distribution of parameters due to variability provides a larger space of parameters for

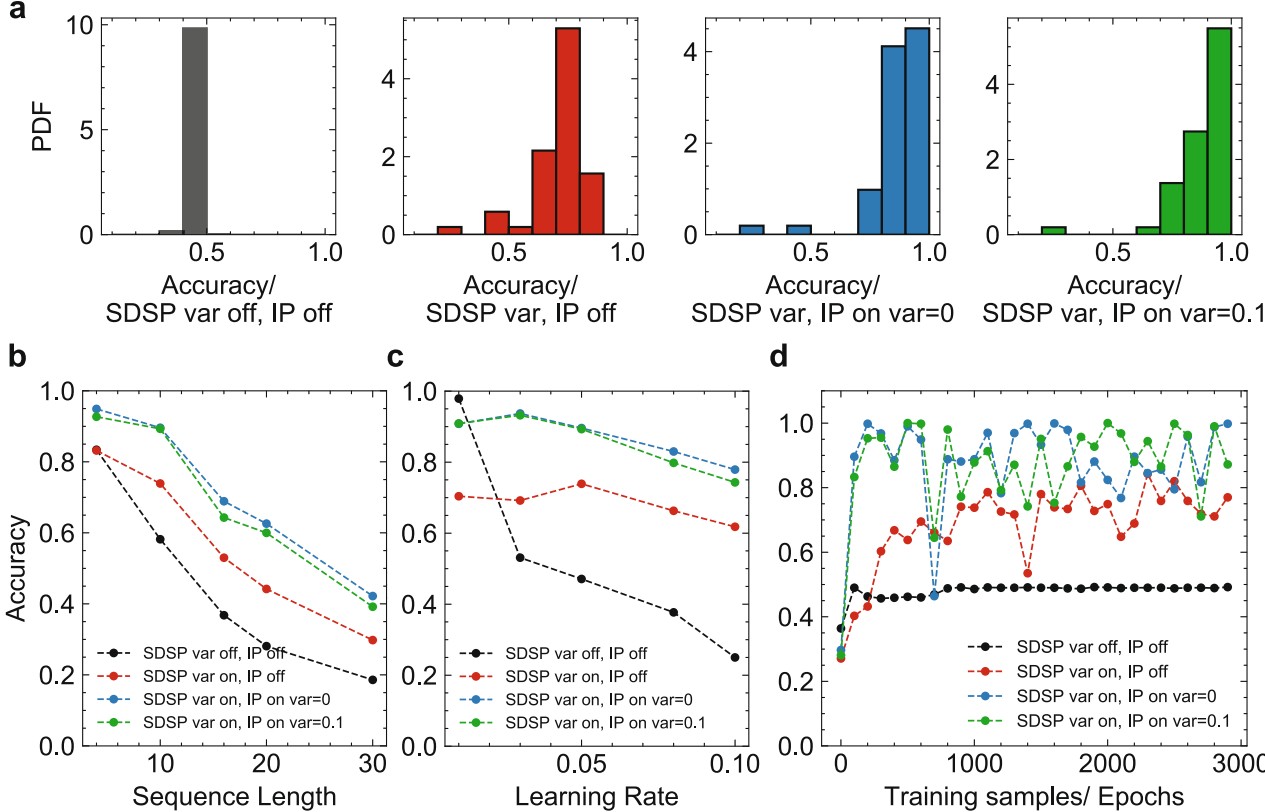

**Fig. 5 | Performance of our proposed self-organized network (MEMSORN) under four different cases of variability in the device models: (i) No variability in devices, and IP operation is off (black), (ii) Variability in devices receiving the SDSP, and IP operation is off (red), (iii) Variability in devices receiving the SDSP, and IP operation is on without variability (Blue), and (iv) Variability in both SDSP and IP learning with standard deviation for the IP devices set to 0.1, taken from our measurements (green). a** Histogram of accuracy for 500 networks confirms the higher accuracy for the networks that include variability and IP operation compared to the other conditions. **b** Accuracy of the MEMSORN network on the counting task with respect to the sequence length. As the sequence length increases, the network needs to remember increasingly more input symbols in the form of a dynamic state of the network, and thus the accuracy drops. Introducing variability, calibrated on measured data, helps the accuracy of the network as all the cases with variability outperform the case without any variability. **c** The average accuracy of the MEMSORN network (for every 100 samples between 1000 to 5000 training epochs) on the counting task with a length of 10 as a function of different learning rates. Introducing variability makes the network robust to hyper-parameter change. **d** Learning evolution of the network accuracy on the counting task with a sequence length of 10 for the four variability cases. Condition (i) has much less noise but has an overall lower accuracy (less than 40%) than the cases where variability and IP are introduced.

search during learning which helps the network to explore and reach a better set of parameters for the task. We discuss this in more detail in the Discussion section.

## Clustering analysis

To understand the dynamics of the static and MEMSORN networks, we performed clustering analysis on the firing activities of the neurons inside the excitatory pool. Figure 6 shows the result of the clustering analysis. First, we reduced the dimensionality of the neural activity using Principle Component Analysis (PCA) (see Methods). Figure 6a plots the PCA of both network activities in response to 50 sequences of length 10. Temporally adjacent letters in the sequence line up next to each other in the principal component space. This indicates the higher structural richness in MEMSORN compared to the static network. Moreover, this helps with the classification accuracy in the readout layer, since the sequences become more linearly separable as indicated by the PCA plot. Figure 6b plots the histogram of explained variance in the firing rate of the random and MEMSORN networks with respect to the first 20 principal components. The explained variance is about 11% more in MEMSORN network compared to the random network suggesting more orderly dynamics in MEMSORN network.

Additionally, we performed a hierarchical clustering analysis on the activity of the SRNN which reveals the formed cell assemblies (see Methods). The result is indicated by the dendrogram in Fig. 6c, showing an increase in the number of uncorrelated clusters in MEM-SORN network compared to the random network. This is the result of more structure emerging from the learning in the recurrent network which is in agreement with the unsupervised memory formation in cell assemblies as we argued in the introduction.

## Discussion

Following a device-algorithm co-design approach, we presented MEMSORN, a hardware architecture that uses its intrinsic properties to self-organize and learn a sequence prediction task. We used a hybrid CMOS/RRAM technology as our hardware substrate, and presented experimental results from the implementation of neurons and synapses in this technology. We then used these statistical measurements to derive "technologically plausible" local learning rules which give rise to self-organization in an SRNN. The self-organization proved to improve the accuracy of the SRNN compared to a fully random network, by more than 15% on a sequence prediction task.

To further investigate the capacity of the network to retain the memory of the past stimuli within its activity, we performed the RANDx4 task proposed in[37] (Methods). Four symbols are randomly presented to the network at different time lags, and the network is to classify the type of signal and its time of presentation at the readout. We have compared the accuracy of the network in four cases with no plasticity, SDSP and IP plasticity applied separately and applied

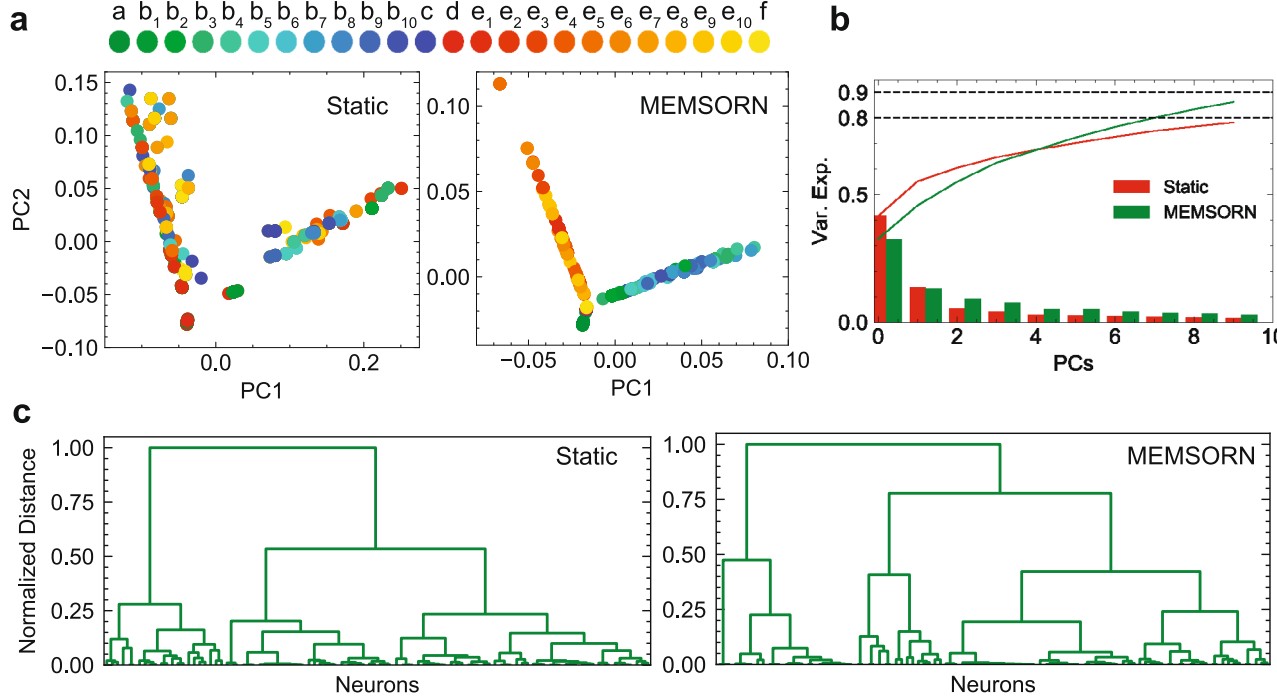

**Fig. 6 | Clustering analysis on the spiking activity of the networks for static and MEMSORN architectures on the counting task with a sequence length of 10.** **a** Principal Component Analysis (PCA) was applied to the firing rate of the two networks in response to 50 sequences of length 10 (600 letters). Each color is assigned to a different position of the letter in the sequence with similar colors encoding the temporal adjacency of the letters in the sequence. In the Principal Component (PC) space, the different input conditions form random clusters in the static network that are not well separated. On the other hand, in MEMSORN network compact clusters are formed which are well separated for different input conditions. **b** Histogram of the captured variance by the first 20 PCs in the static and MEMSORN networks. The explained variance amounts to 79% for the static network, compared to 87% in the MEMSORN, suggesting more orderly dynamics in MEMSORN network. **c** Dendrogram of static and MEMSORN networks showing the hierarchical relationship between clusters of neurons. The normalized height of the dendrogram indicates the distance between the clusters and the links indicate the order in which the clusters are joined. For any given distance, the number of branch numbers for MEMSORN are larger than those for the static network, indicating that the clusters in MEMSORN are better structured.

together. The results for this task are shown in Fig. S1. As reported in[37], the accuracy of the network drops with increasing time lags. However, the combination of the two plasticity rules outperforms the other cases. The SDSP reinforces the mutual information of the input and the SRNN, making it harder to retrieve information from the past. IP spreads the activation of the recurrent neurons more evenly and improves the memory capacity, despite making the activity response too homogeneous. The combination of IP and SDSP results in the best performance, as the associative (SDSP) and homeostatic (IP) forces in the SRNN are perfectly balanced. The PCA of Fig. S2 also reflects these results showing a more cyclic trajectory in MEMSORN compared to the static network. The cyclic nature helps to keep the information in the system in the form of short-term memory.

The unique property of RRAM, compared to other technologies, which was highlighted in this work is its intrinsic variability. The variability provides a distribution of analog values which equips the learning with a large parameter space. We showed that this variability improved the sequence learning by 30% for different sequence lengths. Moreover, introducing the variability while learning, de-sensitized the network to the hyper-parameters, specifically the learning rate.

The positive effect of variability can be explained through the interplay between the SDSP and IP, thoroughly investigated in ref. 37. SDSP learns the structure of the spatio-temporal input transition, by increasing the correlation of the network activity with the input sequence, thus increasing their mutual information. On the other hand, IP increases the neural code space, by ensuring all neurons are part of the computation, thus increasing the redundancy and entropy of the network. Together, synaptic and homeostatic plasticity

cooperate to generate stimulus-sensitive attractors and redundant representations around them. These stimulus-sensitive components are pulled apart by the stimulus-insensitive dynamics, leading to the preservation of the input structure, while separating the neural representations. Adding noise to this system through device variability increases the network state entropy, which helps escape the input-sensitive attractors. This leads to higher accuracy for systems with larger noise as a function of sequence length, learning rate, or epochs as we showed in Fig. 5. Specifically, the device-to-device variability is responsible for decreasing the correlation between the intrinsic attractors in the recurrent network, resulting in increasing the neural code space. On the other hand, the cycle-to-cycle variability of the devices, used in the design of the neurons, provides a search space for finding the optimal neural parameter in order to keep the neural activity in the desired range. This ensures the participation of all the neurons in the computation, further increasing the neural code space. Both sources of variability in the RRAM are present for 'free'. Implementing such randomness using digital circuitry requires bulky circuits such as Linear-feedback shift registers, which use and calculate an algorithm to generate pseudo-random numbers.

Therefore, our approach of "technological plausibility" paves the way for building systems that are potentially more area and also power efficient, as their physical structure gives rise to their function. A technologically-plausible co-design approach closes the gap between the ideas inspired by neuroscience and their applications, algorithms, circuits, and devices. Taking the physics of the devices into account, we designed algorithms that match and exploit them. In our previous works, we have designed CMOS circuits that implement both Hebbian (SDSP) and Homeostatic (IP) plasticity algorithms, shown in Fig. 3d, e.

Synaptic plasticity, i.e., SDSP, circuits were implemented by reading the membrane potential of the post-synaptic neuron on the arrival of the pre-synaptic spike and increasing/decreasing the conductance of the synapse if the sampled value is higher/lower than a reference value[7]. Neuronal plasticity, IP, circuits were implemented by comparing the running average of the neuron's spiking activity to two thresholds which determine the desired range of activity of the neuron[31,35].

These previous works demonstrate the feasibility of the implementation of both learning algorithms using CMOS circuits. In this work, we are including silicon results from our synapse and hybrid neuron implementation, along with system-level evaluation, using simulations that take into account the measurements of the circuit behavior.

### Energy and latency estimations

The MEMSORN hardware reaches convergence for the sequence learning task within the presentation of 1000 learning samples (Fig. 5d), giving rise to ≈ 13 min for learning the task. The power consumption of the MEMSORN's recurrent network during this task consists of three components of static power consumption, dynamic power as a result of the firing of the neurons, and dynamic power due to changing the state of the RRAM devices for learning. The static power consumption of the recurrent layer, including the 1T-1R array and the 200 neurons is 0.2 $\mu$W. The dynamic power due to the firing of the neurons is about 0.8 $\mu$W, and the dynamic power of changing the state of the RRAMs due to IP and SDSP learning is 0.2 $\mu$W (methods). For the duration of the learning, this gives rise to 936 $\mu$J of energy consumption. These values are well within the energy budget and real-time online learning requirement of edge devices[38].

### Comparison to other neuromorphic self-organizing networks

Unsupervised self-organizing networks have been previously studied on different hardware substrates, such as Self-organizing Maps (SOM) on Field Programmable Gate Array (FPGA)[39], and reservoir computing using nano-wire networks[40,41].

The FPGA substrate was used to implement SOMs for multi-modal sensory processing[39]. Hebbian learning was employed to create the SOM in each map, by calculating the weight update based on the input signal. The difference between this implementation and MEMSORN is two-fold; (i) the learning in the SOM network is offline and is learned based on batches of data that are given to the network during learning. In contrast, MEMSORN learns in a fully online fashion, as the input is being streamed, making it suitable for edge computing applications, embedded next to the sensors. (ii) The SOM weight update is calculated by digital hardware, whereas the physics of the MEMSORN substrate naturally gives rise to the weight update and structural changes in the network.

The nano-wire networks are more similar to MEMSORN as they use the physics of randomly dispersed nano-wires and bottom-up self-assembly to create critical recurrent dynamics. The dense unorganized network of these nanowires gives rise to non-linear dynamics creating a reservoir network. Therefore, the nano-wire networks are similar to the static SRNN in our work. Although the principle of creating random connectivity is different, the concept of using randomness to project the input to a high-dimensional space is similar to our static network. As we have demonstrated, MEMSORN with combined plasticity rules using different time scales outperforms the static reservoir framework.

These self-organizing networks are applicable in scenarios where no global teaching signal is available for learning. In the presence of a teaching signal, a supervised learning framework can be used. For online and on-chip learning systems using events, this can be done through approximations of Backpropagation Through Time[42–44], which can be implemented both on digital[45], or in-memory memristive neuromorphic hardware[46]. However, in the absence of supervision,

MEMSORN-like hardware changes its structure and self-organizes to cluster the input signal. MEMSORN implementation provides an architecture where the algorithm is run using the physics of the substrate to adapt to the input stream at the edge.

### Future perspective

MEMSORN takes advantage of the internal properties and statistics of emerging memory technologies for learning. Specifically, the wide distributions of resistive memory in the high-resistive state give rise to a random search algorithm that implements IP. This unguided search can be improved by implementing probabilistic techniques such as Simulated Annealing which reduces the temperature (in this case variability) of the search as the network is converging. This can be achieved by reducing the SET voltage and thus SET probability (IP in Fig. 3d) and thus guiding the search. Moreover, stop-learning criteria can also be added to the SDSP rule as it is done in[47]. Both of these schemes will further reduce the number of device updates and hence energy consumption. We envision a future where intelligent chips enabled by MEMSORN-like designs receive sensory information from the environment and self-organize themselves to interact smoothly with it. This work is a step in that direction.

## Methods
### Fabrication/integration

The circuits of Fig. 1 have been taped-out in 130 nm technology at CEA-Leti, in a 200 mm production line. The Front End of the Line, up to metal 4, has been realized by ST-Microelectronics, while from Metal 5 upwards, including the deposition of the composites for RRAM devices, the process has been completed by CEA Leti. RRAM devices are composed of a 5 nm thick $HfO_2$ layer sandwiched by two 5 nm thick $TiN$ electrodes, forming an $TiN/HfO_2/Ti/TiN$ stack. Each device is accessed by a transistor composing the 1T-1R unit cell. The size of the access transistor is 650 $nm$ in width. 1T-1R cells are integrated with CMOS-based circuits by stacking the RRAM cells on the higher metal layers.

### Device Measurements

For programming and reading the RRAM devices, Source Measure Units (SMU)s from the 4200 SCS Keithley machine were used. We performed statistical analysis from the switching characteristics of a 4 kb array of $HfO_2$-based RRAM.

A SET/RESET operation is performed by applying a positive/negative pulse across the device which forms/disrupts a conductive filament in the memory cell, thus decreasing/increasing its resistance. When the filament is formed, the cell is in the LRS, otherwise the cell is is the HRS. For a SET operation, the bottom of the 1T1R structure (columns in Fig. 1b) is conventionally left at ground level, and a positive voltage is applied to the 1T1R top electrode (rows in Fig. 1b). The reverse is applied in the RESET operation. Typical values for the SET operation are $V_{gate}$ in [0.9 − 1.3] $V$, while the $V_{top}$ peak voltage is normally at 2.0 $V$. For the RESET operation, the gate voltage is instead in the [2.75, 3.25] $V$ range, while the bottom electrode is reaching a peak at 3.0 $V$. The reading operation is performed by limiting the $V_{top}$ voltage to 0.3 $V$, a value that avoids read disturbances, while opening the gate voltage at 4.5 $V$.

**SET and RESET statistics.** To ensure the resistive switching is deterministic, i.e., the device definitely makes the transition from HRS to LRS, a strong $V_{SET}$ is usually applied across the device. If the programming voltage is lowered, a sub-threshold SET is obtained, which makes the switching operation stochastic. We analyzed the sub-threshold SET operation on a population of 4096 1 transistor- 1 RRAM (1T-1R) devices by applying a different range of $V_{SET}$ voltages for 100 cycles, while setting the gate of the transistor to 1.7 $V$. To perform a RESET operation, an opposite voltage $V_{RESET}$ with respect to the SET operation is applied and the gate is biased with $V_{gate}$. HRS of the

**Table 1 | Parameter values for the initialization of the SRNN**

| Neurons | | | Synapses | | SRNN | |
|---|---|---|---|---|---|---|
| | Excitatory | Inhibitory | | | | |
| Number of neurons | 160 | 40 | $\tau$ | 1 ms | $p_{EE}$ | 2% |
| $R_2$ | trained by IP | 1 GΩ | *Weight* | (trained by SDSP) | $p_{II}$ | 0% |
| $R_1$ | 400 MΩ | 600 MΩ | | | $p_{EI}$ | 2% |
| $C_1$ | 10 pF | 10 pF | | | $p_{IE}$ | 10% |
| $\tau_{Ca}$ | 100 ms | 100 ms | | | | |
| $V_{th}$ | 0.2 V | 0.2 V | | | | |
| $R_3$ | 1 GΩ | 1 GΩ | | | | |
| $C_2$ | 2 pF | 2 pF | | | | |
| **IP** | | | **SDSP** | | | |
| $F_T$ | 50 Hz | | $V_{th}$ | 0.2 V | | |
| $\sigma$ | 15 Hz | | $V_{\theta}$ | 0.1 V | | |
| $V_{RESET}$ | 2 V | | *Learning Rate* | 0.01–0.1 | | |

Such values are defined with small-to-absent tuning, with the only aim to guarantee a minimal activation of the network, so to fully rely on the plasticity mechanisms (SDSP and IP) to improve performance. Some parameters, such as the magnitude of RRAM resistance in HRS and the Membrane Capacitance, are forced by technological constraints.

devices after the RESET operation for different $V_{RESET}$ and gate voltages $V_{gate}$ were recorded over 100 cycles. Based on these measurements, we derived the statistical model for the stochastic SET and RESET which were used in our simulations.

### Technologically plausible self-organizing network

**Static network.** The static fully-random network is similar to the Liquid State Machine (LSM)[48] where two populations of excitatory (E) and inhibitory (I) neurons are randomly connected. The excitatory/inhibitory neurons increase/decrease the post-synaptic potential of the neurons to which they are connected. The SRNN neuron population is divided into 80% of excitatory and 20% of inhibitory neurons[48]. An input layer encodes the input information by means of Poisson spike trains. The choice of a noisy input is common practice for Spiking Neural Networks (SNNs) and is justified by the noise-resilient nature of SNNs and stimulation of plasticity mechanisms. The input is randomly projected to the SRNN with a probability of $p_{in}$, set at 0.2 in this work. The excitatory population connects to itself with a probability of $p_{EE}$, and connects to the inhibitory population with the probability $p_{EI}$. The inhibitory population connects to the excitatory population with probability $p_{EI}$. The SRNN represents a complex dynamical system governed by many parameters. In detail, neurons have three parameters (gain, time-constant and threshold), and synapses have 2 parameters (time constant and weight). Table 1 shows the parameter values for the initialization of the SRNN. The output from each of the neurons in the excitatory pool is projected to the output layer, constituted of neurons that encode the output of the network.

**MEMSORN.** MEMSORN is equipped with the two technologically plausible learning rules of SDSP and IP. These local learning rules are only applied to the excitatory neurons and EE connections.

**Modified SDSP.** The measure of correlation in SDSP is the difference between the membrane potential of the post-synaptic neuron $V_{mem}$ to a defined threshold, $V_{\theta}$ at the time of the pre-synaptic spike $t_{pre}$. The weight update on $t_{pre}$ is defined as:

$$w_{EE} = \begin{cases} w_{EE} + LR, & \text{if } V_{mem} \geq V_{\theta} \\ w_{EE} - LR, & \text{otherwise} \end{cases}$$

Where $w_{EE}$ is the weight between the excitatory neurons, and *LR* is the learning rate. The SDSP rule is thus controlled by two parameters, the thresholds applied to the post-synaptic neuron membrane voltage ($V_{\theta}$), and the synaptic weight increment (LR). These values are reported in table 1.

**Intrinsic plasticity.** In SRNNs equipped with plasticity mechanisms, Hebbian synaptic plasticity is thought to create clusters of tightly-bonded neurons, known as attractors. In these networks, IP controls the growth of such attractors and in turn limits the dynamics of neural microcircuits. This effectively improves the information transfer across the SRNN and eventually to the output.

**Algorithm 1.** IP algorithm

$$\begin{aligned}
&\text{Initialization: } R = RESET_{init}(V_{Reset}) \\
&\textbf{while } t < taskDuration \textbf{ do} \\
&\quad \textbf{for } Neurons\ in\ the\ excitatory\ pool \textbf{ do} \\
&\quad\quad \textbf{if } @t_{post}\colon |f_{neuron} - f_T| > \sigma/2 \textbf{ then} \\
&\quad\quad\quad \#\ Sub\text{-}threshold\ Stochastic\ SET \\
&\quad\quad\quad V_{set} = f(|f_{neuron} - f_T|) \\
&\quad\quad\quad p_{set} = P\_subthSET(V_{Set}) \\
&\quad\quad\quad \textbf{if } R_{final} < 50\,k\Omega \textbf{ then} \\
&\quad\quad\quad\quad \#\ RESET \\
&\quad\quad\quad\quad R_{HRS} = RESET()
\end{aligned}$$

**Technologically-Plausible IP.** Algorithm 1 describes the technologically plausible IP algorithm to change the conductance of RRAMs in order to keep the firing rate in a healthy regime. A target firing frequency $f_T$ with an error margin $\sigma$ is defined as the desired range, and the neuron measures its firing rate $f_n$ with respect to the boundaries $f_T \pm \sigma/2$. If $f_n$ moves outside of these boundaries, the value of HRS needs to be updated. To do so, the RRAM is SET with a subthreshold SET voltage which is proportional to the difference between the target and neuron activity. The higher the difference, the higher the SET voltage and thus the higher the probability of setting the device. If the device is SET (i.e., the final resistance $R_{final} < 50\,k\Omega$), we then RESET the device to sample from its internal distribution and find a new value that sets the time constant and gain of the neuron.

IP rule is thus controlled by three parameters of up and down thresholds applied to the neuron's firing rate, and the RESET voltage). The values that are used for all the variables in the learning algorithms are in table 1.

### Sequence presentation and learning

Two patterns of $S_{1n}$ and $S_{2n}$ are presented to the network in random order and sequentially. Each symbol is assigned to a sub-population in the excitatory pool and upon presenting the letter, the corresponding sub-population is stimulated with Poisson spike trains with a frequency of 1 kHz. Each symbol is presented for 50 ms. Once each sequence is completely presented, we wait for 200 ms before presenting the next sequence. This will ensure the activity of the network is decayed away before the next sequence is presented.

**Static network.** The values for the neuron and synapses are initialized and fixed at the beginning of the simulation as is shown in Fig. 3b, c. The neuron parameters are sampled from the HRS distribution fixed on $V_{RESET} = 2\,V$ and the synapse parameters are generated randomly based on $p_{in}$, $p_{EE}$, $p_{EI}$ and $p_{IE}$. The parameters of the static network follow the values in table 1. These hyper-parameters are manually tuned to put the SRNN in an optimal condition.

A general optimum operational condition for SRNNs is the Edge of Chaos, in which the dynamics of the network are neither chaotic nor

deterministic. Traditional approaches to achieve these optimal conditions are either exploiting genetic algorithms[49] or manual hyper-parameter tuning. Random initialization of the weights in the SRNN does not guarantee the formation of multiple clusters of connected neurons, proved to be crucial for memory formation in recurrent networks[2].

**MEMSORN.** Training of the MEMSORN is performed in two steps: first, a purely unsupervised phase in which the network is exposed to inputs, and the two technologically-plausible rules of SDSP and IP shape the activity and response of the SRNN; second, the recurrent weights and neurons' RRAM states are frozen and the output is trained with logistic regression.

In the first phase, the input patterns are presented and synapses modify their weights according to the SDSP rule, creating clusters of neurons that respond to particular spatio-temporal input sequences. The input signal is converted into the activation of different groups of neurons inside the recurrent network. In turn, these sub-groups of neurons are connected to the rest of the SRNN in a random and sparse manner. This results in each input exciting the SRNN in a different way; SDSP is thus capable of exploiting these correlations between parts of the SRNN to reinforce certain connections and form different clusters in the network. At the same time, IP adapts the excitability of neurons in order to control their activity. This prevents strong clusters to take over the SRNN and form a single big group of hyperactive or inactive neurons. Therefore, the benefit of IP is to steer away the SRNN from a regime in which either most neurons are not present in information processing, or they have high output frequency and thus high energy consumption. In MEMSORN, the hyper-parameter tuning of the static network is substituted with the described unsupervised phase. The plasticity mechanisms are exploited in order to find a good enough set of parameters to optimize the performance of the SRNN, alleviating the need to carefully tune or learn the network parameters. These plasticity mechanisms are capable of finding a suitable SRNN configuration depending on the presented input which tunes the network accordingly.

Once the SRNN is tuned, the second phase begins. The activity of the neurons in the network is low-pass filtered through the calcium dynamics of the neuron circuit ($\tau_{Ca}$), indicative of the running average activity. This running firing rate is registered at the end of the sequence and utilized in a logistic regression algorithm to calculate the output weights in the readout.

The logistic regression could also be replaced with the online delta rule in an always-on fashion[50]. Such circuit implementation of the delta rule algorithm allows training the output layer using Stochastic Gradient Descent (SGD) in a one-layer SNN. This kind of system will continuously adapt to the input stimuli in a low power always-on manner.

### Latency and energy calculations

As is shown in Fig. 5d, the learning takes about 1000 presentations of samples to converge. Each symbol is presented for 50 ms with a 200 ms of wait time in between pattern presentations. For a pattern of 12 symbols ($n = 10$), that gives rise to 800 ms per epoch. Therefore, the total latency is 800 s or 13 min.

We identified three sources of power consumption in our system: static power, dynamic power due to the firing of the neurons, and dynamic power due to the state of the RRAM devices changing.

**Static power.** The static power consumption of each neuron, together with the switches is 1.4 nW. This gives rise to $\approx 0.2\,\mu W$ of static power consumption for the entire population of the neurons.

**Neuron dynamic power.** Based on our measurements, the energy consumption of our neuron in 130 nm process is 100 pJ/spike. Our recurrent layer has 160 excitatory neurons firing at 50 Hz (maintained by the IP algorithm). This gives rise to:

$$160 \times 50\ spikes/second \times 100\ pJ/spike \approx 0.8\mu W$$

**RRAM state change dynamic power.** During the learning operation, the state of the RRAM devices changes due to the IP and SDSP learning rules. We have counted the total number of times that the state of the devices is changing during the learning process, which is $\approx 3 \times 10^6$ times. As we have previously reported[31], each RRAM SET and RESET cycle consumes around 50 pJ. For the duration of the learning (13 mins), this gives rise to $50\,pJ \times 3 \times 10^6/(13 \times 60\,s) = 0.2\,\mu W$.

Therefore, for the duration of the learning process, the total energy consumption is

$$(0.2\mu W + 0.8\mu W + 0.2\mu W) \times 13 \times 60\ seconds = 1.2\,\mu W \times 780\ seconds = 936\,\mu J.$$

### RANDx4 task

As is detailed in[37], in RANDx4 task, the recurrent network is driven by four randomly drawn inputs $P = A, B, C, D$. The receptive fields of non-overlapping subsets of neurons are tuned exclusively to each input. In our case, the 100 neurons reservoir was split into the excitatory (80% of the total) and inhibitory (20%) populations. Input stimuli are directed towards small receptive fields of 12 excitatory neurons in the reservoir. Inputs are encoded as Poisson pulse trains at the average frequency of 5 kHz, each activated when the corresponding letter $A$, $B, C, D$ is randomly chosen.

We have performed the RANDx4 task by setting the duration of each input stimuli approximately the same as that of the silicon neurons and synapses, at 1 ms. This assures that the memory capacity of the network does not stem from the components alone - neurons and dynamical synapses - and it rather leverages the organization at the network level.

### Clustering Analysis

We show 50 sequences of length 12 ($n = 10$, 600 letters) to the static and MEMSORN networks and record the spike times of all the excitatory populations for each shown symbol. This spiking data is then low pass filtered with a time constant of $\tau = 5\,ms$. We take the data point at the last time step of the symbol presentation for all the symbols in the two sequences during 50 data presentations.

**PCA.** We reduce the dimension of the SRNN activity from 160 excitatory neurons to the first two Principle Components (PCs) and plot the first and second PCs in time. This is plotted in Fig. 6a.

**Hierarchical clustering.** We also analyze the obtained SRNN activity using hierarchical clustering analysis. This is a type of unsupervised learning algorithm used to cluster the data points with similar characteristics. We use the agglomerative hierarchical clustering which is a "bottom-up" approach where each data point starts in its own cluster. Moving up the hierarchy, clusters are formed by joining the two closest data points resulting in increasingly fewer clusters with higher distance or dissimilarity. This hierarchy of clusters is represented as a tree (or dendrogram). The root of the tree is the unique cluster that gathers all the samples, the leaves being the clusters with only one sample. For performing this analysis, we have used the "cluster.hierarchy.linkage" function from the python scipy library. We have calculated the distance between the clusters with the "ward" method which minimizes the total within-cluster variance. At each step, this method merges pairs of clusters which lead to a minimum increase in the total within-cluster variance after merging. This increase is a weighted squared distance between cluster centers.

## Data availability

Data are not publicly available for commercial sensitivity. This work was done in collaboration with two industrial partners (CEA-LETI, France, and Toshiba, Japan). Therefore, the data can only be shared upon request as part of the company's policies.

## Code availability

Code is not publicly available for commercial sensitivity. This work was done in collaboration with two industrial partners (CEA-LETI, France, and Toshiba, Japan). Therefore, the code can only be shared upon request as part of the company's policies.

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

## Acknowledgements
This work is partly supported by Toshiba corporation, EU Horizon 2020 Memscales project (871371), and the SNSF grant SMALL (20CH21_186999). We are grateful to Stefano Brivio from the National Research Counsil (CNR), Italy, and Christian Tetzlaff from the University of Gottingen, Germany for their critical and valuable comments on the manuscript.

## Author contributions
M.P. proposed the idea. M.P., F.M., and T.D. conceived the experiment(s). F.M. and K.N. conducted the software experiment(s). M.P, F.M., K.N., Y.N., and G.I. analyzed the data. M.P and T.D. designed the circuits. The circuit was fabricated at CEA-LETI under the supervision of E.V.; F.M. performed the measurement on the fabricated chips under the supervision of E.V. M.P., F.M. and G.I. wrote the manuscript. All authors reviewed the manuscript.

## Competing interests
The authors declare no competing interests.
