## [Peer Review File · Nature Communications]

Title: Self-organization of an inhomogeneous memristive hardware for sequence learningREVIEWER COMMENTS

Reviewer #2 (Remarks to the Author):

As a matter of fairness and transparency I always reveal my identity for a review process. My name is Gordon Pipa, and I enjoyed reading your manuscript. Overall it is clearly written and presents a new and highly relevant technological development. Given my expertise on computational principles and machine learning I will mainly focus on the evaluation of the non-electrical engineering parts. While I cannot judge the latter parts from a technical perspective in detail, I would like to stress that the paper was written in a way that I could follow also these more technical parts really well. Having said that, I would like to emphasize that the paper has a good readability in general and is suited for a more general audience.

From the point of view of a computational device the paper presents a very relevant and highly interesting implementation of a self-organized neuronal network. It closely resembles a rather principle model of self-organization (SORN) using the technology of Memristive elements. Neurons are implemented using a hybrid CMOS/RRAM design. The Memory is used to store the parameters of the neurons, such as gain and refractory period. Figure 2(a-c) show the technical properties with remarkable variability that is addressed in the paper in the later parts.

The benchmarks are based on a task from the original SORN paper. The task was tailored for testing memory. Figure two shows the performance of the system based on an output prediction. Figure 2b is not fully clear. I guess a usual presentation of the results would be based on a confusion matrix, illustrating the probability output classes in respect to the expected output. The unusual simplification chosen in figure 2b remains unclear to me and should be changed in case there isn't a factor that I haven't understood so far. I also do not understand the information from the figure legend "The missing output in MEMSORN is due to the uncertainty about the next sequence as it is randomly chosen". I can neither identify the link to the figure, nor do I understand what is meant here.

The section on the effect of variability/noise is rather descriptive. While most results can be read as support for idea that the inclusion of plasticity is boosting the performance, the effects underlying this boosting are not sufficiently discussed. Maybe the discussion of this paper would be a good place to do this. Especially the effect of the different sources of noise in this technology are not discussed to a sufficient degree. Maybe it helps to link to related work that explains the improvements of IP and synaptic plasticity in the presence of noise for the original SORN

<https://doi.org/10.1371/journal.pcbi.1003512>. This work explains the effect of IP and relates it to an increase in entropy that increases the coding spaces, while the synaptic plasticity was linked to increased mutual information between input and network response. These observations seem to rather closely resembled by your PCA and cluster analysis.

In general the characterization of a computational device requires more than the features of the counting task of the original SORN paper. For example it would be interesting to see whether the system can react on new input that requires the response to "switch" between trajectories that are learned by the dynamical system. A task that can be used for that would be RANDx4, Markov-85 from <https://doi.org/10.1371/journal.pcbi.1003512>. While a focus on a single test would be a strong limitation for a paper in general, it is sufficient in the context of this paper more technology focused

submission. In that sense the example computation based on the counting task is sufficient to prove the computational features claimed by the paper.

In general I was surprised to see no discussion of the speed and energy consumption of the system. Isn't exactly this the unique selling proposition of a neuromorphic implementation? What is the current status and future perspective of the system? Are there limitations of the current system in that respect? How does this system compare to other implementations?

Overall, this is a very interesting, innovative and cutting edge research and technology presented here. The submission can and should be improved by clarifying the above points. In case that there are space limitation I felt that the Clustering section could be shorted significantly and fused with the previous section. The detailed procedure of the cluster can be moved to the appendix. This additional space shall be invested for a longer and more detailed discussion. I would be happy to see the submission being improved and resubmitted. All the best and success, Gordon

Reviewer #3 (Remarks to the Author):

The authors present "MEMSORN," a mapping of the SORN self-organizing recurrent neural network [3] to a memristive (RRAM) hardware substrate. The chief result of the work is to replicate the approach and the experiments of SORN on their custom neuromorphic RRAM hardware (130nm CMOS + HfO₂ RRAM). The same result as the original 2009 SORN paper [3], they show that the learning in the self-organizing network enables the network to outperform a static (non-learning) reservoir network on the same sequence task. The network dynamics and learning rules match the SORN work.

The chief value of the work, however, is in demonstrating a potential approach for overcoming the variability in RRAM based neuromorphic systems. Process variability in manufactured RRAM devices, as well as variability in reliably and accurately programming the devices severely curtails their use outside of research labs. If networks cannot be accurately programmed on a system, the system behavior and performance cannot be accurately predicted, let alone guaranteed. In this work, the hardware is adaptively programmed using the learning rules, to initialize the system to an operational state (with a certain probability). Integrated adaptive learning presents a potentially feasible approach to adjust for the stochastic variability in these systems.

Issues with the work:

1. The Hebbian/Homeostatic plasticity learning rules appear to be implemented off-chip. The hardware itself is adaptable -- the RRAM devices can be adjusted. However, the learning algorithms (e.g. algorithm 1) are implemented off chip. Thus, the hardware is not a fully self-contained adaptive system.

2. What is the best performance for other algorithms on this sequence task (i.e. non-spiking, etc.)? Fig 4. shows MEMSORN outperforms a static reservoir network. But how does MEMSORN compare to other state-of-the-art approaches?

Or how does MEMSORN compare to other state-of-the-art approaches on other common sequence learning tasks?

Response to the reviewers:

MEMSORN: Self-organization of an inhomogeneous memristive hardware for sequence learning

We sincerely thank all the reviewers for their insightful comments on our manuscript. We believe addressing the comments has made the paper much stronger and we are grateful for that. Below, we reply to each concern of the reviewers one by one. We have addressed the changes in the main and supplementary texts, highlighted with a strike-through red when text was removed, and in blue when text was added.

Response to Reviewer 1

As a matter of fairness and transparency I always reveal my identity for a review process. My name is Gordon Pipa, and I enjoyed reading the/your manuscript. Overall it is clearly written and presents a new and highly relevant technological development. Given my expertise on computational principles and machine learning I will mainly focus on the evaluation of the non-electrical engineering parts. While I cannot judge the latter parts from a technical perspective in detail, I would like to stress that the paper was written in a way that I could follow also these more technical parts really well. Having said that, I would like to emphasize that the paper has a good readability in general and is suited for a more general audience.

Reply 1 Dear Gordon, We very much appreciate this transparency and we value it very much for the review process. Thank you for the encouraging comments. It is very encouraging to hear that the paper is written in such a way that is suited for a broad audience.

From the point of view of a computational device the paper presents a very relevant and highly interesting implementation of a self-organized neuronal network. It closely resembles a rather principle model of self-organization (SORN) using the technology of Memristive elements. Neurons are implemented using a hybrid CMOS/RRAM design. The Memory is used to store the parameters of the neurons, such as gain and refractory period. Figure 2(a-c) show the technical properties with remarkable variability that is addressed in the paper in the later parts.

Reply 2 Thank you very much for your comment and for this summary. Although it was not explicitly asked, we have performed additional measurements on our fabricated dies, and included a more accurate measurements of our hybrid CMOS/RRAM neuron design. This is reflected in Fig. 2d and 2e of the main text. Changing R2 (as shown in Fig. 1c) changes the time constant of our hybrid CMOS/RRAM neuron, and that in turn changes the frequency of the neuron's firing. We use this property in our Intrinsic Plasticity design as is described in the paper.

The benchmarks are based on a task from the original SORN paper. The task was tailored for testing memory. Figure two shows the performance of the system based on an output prediction. Figure 2b is not fully clear. I guess a usual presentation of the results would be based on a confusion matrix, illustrating the probability output classes in respect to the expected output. The unusual simplification chosen in figure 2b remains unclear to me and should be changed in case there isn't a factor that I haven't understood so far. I also do not understand the information from the figure legend "The missing output in MEMSORN is due to the uncertainty about the next sequence as it is randomly chosen". I can neither identify the link to the figure, nor do I understand what is meant here.

Reply 3 Reply: Thank you very much for your suggestion. We have changed the representation to confusion matrix as you suggested. The sentence about the "The missing output ..." meant to say that since the order of the input sequences is random, input symbols are equiprobable and their temporal succession carries no structure. Therefore, the accuracy drops to chance level between the two consecutive sequences.

Action: We have changed the accuracy representation to confusion matrix in Fig. 4b, and have rewritten the caption to clarify the unclear sentence about the accuracy of the network between two consecutive sequences.

The section on the effect of variability/noise is rather descriptive. While most results can be read as support for idea that the inclusion of plasticity is boosting the performance, the effects underlying this boosting are not sufficiently discussed. Maybe the discussion of this paper would be a good place to do this. Especially the effect of the different sources of noise in this technology are not discussed to a sufficient degree. Maybe it helps to link to related work that explains the improvements of IP

and synaptic plasticity in the presence of noise for the original SORN <https://doi.org/10.1371/journal.pcbi.1003512>. This work explains the effect of IP and relates it to an increase in entropy that increases the coding spaces, while the synaptic plasticity was linked to increased mutual information between input and network response. These observations seem to rather closely resembled by your PCA and cluster analysis.

Reply 4 Reply: Thank you very much for bringing this interesting paper to our attention. Indeed, the noise analysis of the PLOS paper is highly relevant to our case and it helped us explaining the noise effect better. Especially, we explained the effect of the different sources of variability using the interplay between IP and SDSP.

Action: Inspired by the paper¹ analysis, we have added a paragraph in the discussion section about the interplay between IP and SDSP. We explained how the addition of noise through device variability increases the network state entropy and thus neural code space, which reduces the input-insensitive dynamics, leading to higher accuracy. We have specified the role of different source of variability including device-to-device and cycle-to-cycle variability. Device-to-device variability reduces the correlation between the intrinsic attractors of the system leading to an increased neural code space. Cycle-to-cycle variability of the devices, used in the design of the neurons, increases the search space for finding the optimal neural parameters to keep the activity of the neurons in the desired range of activity which ensures all neurons are part of the computation. This further increase the neural code space. This explanation is reflected in the modified main text, in the discussion section in lines 207-221 of the main text.

In general the characterization of a computational device requires more than the features of the counting task of the original SORN paper. For example it would be interesting to see whether the system can react on new input that requires the response to “switch” between trajectories that are learned by the dynamical system. A task that can be used for that would be RANDx4, Markov-85 from <https://doi.org/10.1371/journal.pcbi.1003512> . While a focus on a single test would be a strong limitation for a paper in general, it is sufficient in the context of this paper more technology focused submission. In that sense the example computation based on the counting task is sufficient to prove the computational features claimed by the paper.

Reply 5 Reply: We appreciate the suggestion of the RANDx4 task as a new benchmark which further highlights the benefit of our proposed plasticity mechanisms. We have performed the RANDx4 task by setting the duration of each input stimuli approximately the same as that of the silicon neurons and synapses, at 1 ms. This ensures that the memory capacity of the network does not stem from the components alone - neurons and dynamical synapses - and it rather leverages the organization at the network level. The result of the RANDx4 task are plotted in Fig. R1 below.

Figure R1. MEMSORN results on the RANDx4 task. Each input letter is randomly selected and presented to the SRNN for an interval of time approximately equivalent to the time constant of the neurons and synapse circuits. The formation of memory thus solely attributes to the formation of a dynamical state mediated by the plasticity mechanisms. SDSP helps the SRNN to separate the response of the different input, but has a negative impact on the memory capacity. IP makes the network response more homogeneous across all neurons, and improves the memory capacity. The combination of SDSP and IP allows to improve the performance, outperforming the static and stand-alone plasticity cases, as demonstrated in earlier works¹.

Similar to¹, the accuracy of the network decreases as the time-lag increases, since the network is requested to retrieve

information increasingly longer from the past. In accordance to Fig. 2b in¹, the Hebbian learning (or STDP in¹) reinforces the mutual information of the input and the SRNN, making it harder to retrieve information from the past. IP spreads the activation of the SRNN more evenly and improves the memory capacity, despite making the network activity response too homogeneous. The combination of IP and SDSP results in the best performance, as the associative (SDSP) and homeostatic (IP) plasticity rules in the SRNN are perfectly balanced. This is further validated by the Principle Component Analysis in Figure R2 below.

Figure R2. RANDx4 task: Principle Component Analysis (PCA) on the activity of the static network (a) and MEMSORN (b). The PCA on the MEMSORN shows a more cyclic trajectory than the static network. The cyclic nature helps to keep the information in the system in the form of short-term memory. The PCA justifies the larger accuracy of the MEMSORN.

Action: We have included the new accuracy and PCA results on the RANDx4 task as a new benchmark in the Figures S1 and S2 of the Supplementary information. We discuss the results in the discussion section which confirms our findings on the counting task: the combination of the two learning algorithms outperforms the static network, and the employment of each plasticity mechanism separately. This part is reflected in lines 192-202 of the main text.

In general I was surprised to see no discussion of the speed and energy consumption of the system. Isn't exactly this the unique selling proposition of a neuromorphic implementation? What is the current status and future perspective of the system? Are there limitations of the current system in that respect? How does this system compare to other implementations?

Reply 6 Indeed, the addition of the estimations on energy and latency is important for the neuromorphic implementation and we have now included the estimations for both in the manuscript. Our estimation is based on the number of epochs that it takes for the network to reach its stable accuracy. As shown in Fig. 5d of the main text, it takes about 1000 learning epochs for the network to reach a stable accuracy on the counting task. Each symbol is presented for 50 ms with a 200 ms of wait time in between pattern presentations. For a pattern of 12 symbols ($n=10$), that gives rise to 800 ms per epoch. Therefore, the total latency for 1000 epochs is 800 seconds, or 13 minutes. To calculate energy consumption, we have calculated the energy consumption of the neuron populations based on their average activity, and the energy consumption due to the programming of the RRAM devices. Given 160 excitatory neurons in MEMSORN with IP-regulated firing activity at 50 Hz and 30 pJ of energy per spike (based on state-of-the-art neuron design²) for the duration of the learning, the neuron's energy consumption is ≈ 200 nJ. Moreover, as we have previously reported³, each RRAM SET and RESET cycle consumes about 50 pJ. We have counted the number of times that SDSP and IP are used in 1000 epochs, giving rise to a total energy of 120 nJ.

For comparison with other works, we have now included all the hardware implementations of self-organized networks, in a newly introduced subsection in the discussion section titled "Comparison to other neuromorphic self-organizing networks". The implementations include more conventional substrates such as Field Programmable Gate Array (FPGA), and more novel substrates such as self-assembled nano-wires.

Finally, the limitations of the system and future perspectives are now added in another newly-introduced subsection titled "Future perspective". In this paper, we are exploiting the variability of the resistive memory to implement a search algorithm for the optimal resistive state of each neuron in the network, such that the activity of the neurons remain in a desirable regime. This unguided search can be improved by implementing probabilistic techniques such as simulated annealing which reduces the searching ability of the network as it gets closer to the solution. This can be implemented by introducing a stop-learning region for the SDSP and reducing the probability of SET during the IP algorithm (Fig. 5d in the main text).

Action: (i) We have now added the energy and latency estimation subsection in the discussion section with more details about the calculations in the methods; (ii) We have added another subsection in the discussion section on "Comparison to other neuromorphic self-organizing networks" and we compared MEMSORN to other implementations of self-organizing networks, e.g., on FPGA or nano-wires; (iii) we have added a subsection on "Future perspective" suggesting how the MEMSORN implementation can be improved. These are reflected in lines 234-271 of the main text.

Overall, this is a very interesting, innovative and cutting edge research and technology presented here. The submission can and should be improved by clarifying the above points. In case that there are space limitation I felt that the Clustering section could be shorted significantly and fused with the previous section. The detailed procedure of the cluster can be moved to the appendix. This additional space shall be invested for a longer and more detailed discussion. I would be happy to see the submission being improved and resubmitted. All the best and success, Gordon

Reply 7 We very much appreciate your constructive comments which we think significantly improved the paper. Following your comments, (i) we have added a new benchmark on RANDx4 task and analyzed the network activity during the task; (ii) we have added a subsection on energy and latency estimations of the MEMSORN network; (iii) we compared MEMSORN to other hardware efforts for implementing self-organizing networks; and (iv) we have added a subsection on limitations and future perspectives of our approach. We hope we have been able to address your concerns and you find this revision suitable for publication in Nature Communications.

Response to Reviewer 2

The authors present "MEMSORN," a mapping of the SORN self-organizing recurrent neural network [3] to a memristive (RRAM) hardware substrate. The chief result of the work is to replicate the approach and the experiments of SORN on their custom neuromorphic RRAM hardware (130nmCMOS + HfO₂RRAM). The same result as the original 2009 SORN paper [3], they show that the learning in the self-organizing network enables the network to outperform a static (non-learning) reservoir network on the same sequence task. The network dynamics and learning rules match the SORN work.

The chief value of the work, however, is in demonstrating a potential approach for overcoming the variability in RRAM based neuromorphic systems. Process variability in manufactured RRAM devices, as well as variability in reliably and accurately programming the devices severely curtails their use outside of research labs. If networks cannot be accurately programmed on a system, the system behavior and performance cannot be accurately predicted, let alone guaranteed. In this work, the hardware is adaptively programmed using the learning rules, to initialize the system to an operational state (with a certain probability). Integrated adaptive learning presents a potentially feasible approach to adjust for the stochastic variability in these systems.

Reply 8 Thank you very much for your summary and pointing out the key results and contribution of our work, especially regarding the variability problem in resistive memory which we have turned into a feature in this work.

1. The Hebbian/Homeostatic plasticity learning rules appear to be implemented off-chip. The hardware itself is adaptable – the RRAM devices can be adjusted. However, the learning algorithms (e.g. algorithm 1) are implemented off chip. Thus, the hardware is not a fully self-contained adaptive system.

Reply 9 Reply: Thank you for your feedback. We realized that in the first submission we did not sufficiently explain this point. Both learning algorithms (SDSP (Hebbian) and IP (Homeostatic)) can be implemented on chip, as we have done before in references³⁻⁵. In⁴ we implemented the circuits for synaptic plasticity, SDSP, by reading the membrane potential of the post-synaptic neuron on the arrival of the pre-synaptic spike, and increase/decrease the conductance of the synapse, if the sampled value is higher/lower than a reference value. In^{3,5}, we implemented the circuits for neuronal plasticity, IP, through comparing the running average of the neuron's spiking activity to two thresholds which determine the desired range of activity

of the neuron. If the neuron goes outside of this range, we send a pulse for a SET-then-RESET operation to the memristive device implementing the time constant of the neuron. This operation is equivalent to sampling from a log-normal distribution and picking a new value for the time constant of the neuron. Therefore, through previous work we have shown the feasibility of the implementation of both learning algorithms using CMOS circuits.

Here, we are including silicon results from our synapse and hybrid neuron implementation, along with system level evaluation using simulations which take into account the device characteristics and circuit behavior. We are reporting simulations results of the self-organization of the system through the learning algorithms which can be implemented on chip.

Action: Although it was not explicitly asked, we have performed additional measurements on our fabricated dies, and included a more accurate measurements of our hybrid CMOS/RRAM neuron design. This is reflected in Fig. 2d and 2e of the main text. Changing R_2 (as shown in Fig. 1c) changes the time constant of our hybrid CMOS/RRAM neuron, and that in turn changes the frequency of the neuron's firing. We use this property in our Intrinsic Plasticity design as is described in the paper. We have clarified the hardware implementation feasibility of the algorithms in the discussion section by pointing to our previous publications. We discuss how the circuits for SDSP and IP can be implemented and highlight the contribution of this work more clearly. This is reflected in lines 222-233.

2. What is the best performance for other algorithms on this sequence task (i.e. non-spiking, etc.)? Fig 4. shows MEMSORN outperforms a static reservoir network. But how does MEMSORN compare to other state-of-the-art approaches? Or how does MEMSORN compare to other state-of-the-art approaches on other common sequence learning tasks?

Reply 10 Reply Thank you very much for your important comment. The conventional non-spiking models such as Long Short Term Memory (LSTM) solve the counting task by overfitting, giving an accuracy of 100%. This is because there are not enough data available for the deep learning model to fit a generalized model to the task. The unique point about the MEMSORN model (and models alike) is three fold: (i) the unsupervised self organization of the network; (ii) its online learning nature; (iii) implementation of the learning which is embedded and derived from the physics of the substrate. We expand these points below:

(i) the clustering of the information in the recurrent network happens in a fully unsupervised manner. The recurrent network self organizes into clusters of neural activity representing spatio-temporal features of the streaming input. This is as a result of the two plasticity mechanisms in different time scales which cooperate to increase the information capacity of the network while increasing the mutual information with the input. (ii) The network does not require a pre-recorded dataset for training in batch mode. Rather, it learns the parameters of the network event by event, in real time, as they arrive to the network. (iii) The hardware implementation *is* the algorithm for learning and it optimally uses the substrate properties to adapt to the input stream. These points make MEMSORN and the networks alike ideal for applications close to the edge of the sensors, known as *edge computing*. At the edge, the MEMSORN network will be able to learn the statistics of the sensory input in real-time as they stream, and change its structure.

Action To compare with the state of the art hardware implementations, we added a new subsection to the discussion section titled "comparison to other neuromorphic self-organizing networks" where we included the neuromorphic self-organized network implementations including the more conventional substrates such as FPGAs, and more novel substrates such as self-assembled nano-wires. We compare and discuss the benefits of the MEMSORN approach to the state of the art and we discuss the unique features and application domain of MEMSORN. These points are addressed in lines 240-262. We have also included a new section "Future perspective" which discusses how MEMSORN can be optimized for the next steps. This is also included in lines 263-271.

Reply 11 We thank the reviewer very much for raising important questions regarding the on-chip implementation and comparison to the state of the art. We hope we have been able to address your concerns by the additional sections we have added in the discussion section.

References

1. Toutounji, H. & Pipa, G. Spatiotemporal computations of an excitable and plastic brain: neuronal plasticity leads to noise-robust and noise-constructive computations. *PLoS computational biology* **10**, e1003512 (2014).

2. Rubino, A., Payvand, M. & Indiveri, G. Ultra-low power silicon neuron circuit for extreme-edge neuromorphic intelligence. In *International Conference on Electronics, Circuits, and Systems, (ICECS), 2019*, 458–461, DOI: [10.1109/ICECS46596.2019.8964713](https://doi.org/10.1109/ICECS46596.2019.8964713) (2019).
3. Dalgaty, T. *et al.* Hybrid CMOS-RRAM neurons with intrinsic plasticity. In *International Symposium on Circuits and Systems (ISCAS), 2019* (IEEE, 2019).
4. Qiao, N. *et al.* A reconfigurable on-line learning spiking neuromorphic processor comprising 256 neurons and 128k synapses. *Front. neuroscience* **9**, 141 (2015).
5. Dalgaty, T. *et al.* Hybrid neuromorphic circuits exploiting non-conventional properties of RRAM for massively parallel local plasticity mechanisms. *APL Mater.* **7**, 081125, DOI: [10.1063/1.5108663](https://doi.org/10.1063/1.5108663) (2019).

REVIEWER COMMENTS

Reviewer #3 (Remarks to the Author):

Issues with the revision:

1) The following statement does not make sense. It is not grounded in the facts presented by the paper:

"MEMSORN implementation provides an architecture where the substrate is the algorithm for learning, and it optimally uses the substrate properties to adapt to the input stream at the edge."

- The substrate of RRAM-based neurons and synapses is not an algorithm. SDSP and IP are algorithms.
- There is no proof of optimality presented in the paper. Wherever "optimal" is used in the paper, it is more accurately described as "feasible."

2) The energy calculation is incorrect.

The Introduction states:

"We design and fabricate the RRAM-based synapse and neurons in 130 nm Complementary Metal-Oxide-Semiconductor (CMOS) technology integrated with HfO₂-based RRAM devices"

The Methods section states:

"The circuits of Fig. 1 have been taped-out in 130 nm technology at CEA-Leti,..."

However, the energy calculation is based on a different process technology! This inconsistency has a dramatic effect on the energy consumption.

"Using the state-of-the-art neuron design in Global Foundry 22nm technology, each neuron consumes around 30pJ at 50Hz and for the duration of the learning, the entire energy consumption will be around 200nJ."

3) Also, the revision could use an overall edit pass to fix grammar issues, such as missing/incorrect articles, sentence fragments, etc.

Response to the reviewers:

MEMSORN: Self-organization of an inhomogeneous memristive hardware for sequence learning

We sincerely thank the reviewer for their detailed comments on our manuscript. Below, we reply to each concern of the reviewer one by one. We have addressed the changes in the main text, highlighted with a red strike-through, when text was removed, and in blue, when text was added.

Response to Reviewer 2

(Gordon) - I thank the authors for this very positive and very goal oriented interaction. All suggestions were addressed and improved the paper in terms of quality and readability. I fully support the paper and recommend acceptance. It is a important step towards using neuromorphic computing based on memristors.

Congrats, Gordon

Reply 1 Thank you very much for accepting our paper and acknowledging its advancement in neuromorphic computing based on memristive devices. We thank you for the comments and feedback which improved the paper significantly.

Response to Reviewer 3

Issues with the revision:

1) The following statement does not make sense. It is not grounded in the facts presented by the paper:

"MEMSORN implementation provides an architecture where the substrate is the algorithm for learning, and it optimally uses the substrate properties to adapt to the input stream at the edge."

- The substrate of RRAM-based neurons and synapses is not an algorithm. SDSP and IP are algorithms. - There is no proof of optimality presented in the paper. Wherever "optimal" is used in the paper, it is more accurately described as "feasible."

Reply 2

Reply and action: We changed the sentence to the following to remove the ambiguity:

"MEMSORN implementation provides an architecture where the algorithm is run using the physics of the substrate to adapt to the input stream at the edge. "

This is reflected in line 266 of the main text.

2) The energy calculation is incorrect.

The Introduction states: "We design and fabricate the RRAM-based synapse and neurons in 130 nm Complementary Metal-Oxide-Semiconductor (CMOS) technology integrated with HfO₂-based RRAM devices"

The Methods section states: "The circuits of Fig. 1 have been taped-out in 130 nm technology at CEA-Leti,..."

However, the energy calculation is based on a different process technology! This inconsistency has a dramatic effect on the energy consumption. "Using the state-of-the-art neuron design in Global Foundry 22nm technology, each neuron consumes around 30pJ at 50Hz and for the duration of the learning, the entire energy consumption will be around 200nJ."

Reply 3

Reply Thank you for pointing this out. We had used the energy consumption of a neuron designed in 22 nm technology, which was optimized for energy. But even with the 22 nm design, we had made a mistake with the calculations.

We identified three sources of power consumption in our system: static power, dynamic power due to neuron firing, dynamic power due to the state of the RRAM devices changing.

Static power: The static power consumption of each neuron, together with all the switches on the path is 1.4 nW. This gives rise to $\approx 0.2 \mu\text{W}$ of static power consumption for the entire population of the neurons.

Neuron dynamic power Based on our measurements, the energy consumption of our neuron in 130 nm process is 100 pJ/spike. Our recurrent layer has 160 excitatory neurons firing at 50 Hz (maintained by the IP algorithm). This gives rise to: $160 \times 50 \text{ spikes/second} \times 100 \text{ pJ/spike} \approx 0.8 \mu\text{W}$

RRAM state change dynamic power During the learning operation, the state of the RRAM devices changes due to the IP and SDSF learning rules. We have counted the total number of times that the state of the devices are changing during the learning process, which is $\approx 3 \times 10^6$ times. As we have previously reported¹, each RRAM SET and RESET cycle consumes around 50 pJ. For the duration of the learning (13 minutes), this gives rise to:

$$50 \text{ pJ} \times 3 \times 10^6 / (13 \times 60 \text{ seconds}) = 0.2 \mu\text{W}.$$

Therefore, for the duration of the learning process, the total energy consumption is:

$$(0.2 \mu\text{W} + 0.8 \mu\text{W} + 0.2 \mu\text{W}) \times 13 \times 60 \text{ seconds} = 1.2 \mu\text{W} \times 780 \text{ seconds} = 936 \mu\text{J}.$$

Action These changes are now reflected both in “Energy and latency estimation” subsection in the main text (lines 236-244 of the main text), and in the Methods section (lines 373 to 390 of the main text).

3) Also, the revision could use an overall edit pass to fix grammar issues, such as missing/incorrect articles, sentence fragments, etc.

Reply 4 Thank you very much for pointing out the grammar issues with the paper. We have gone through it carefully and have addressed the issues to the best of our ability. You can see the changes in the diff.pdf file attached to the revision submission.

Reply 5 We thank you again for your time for going through our paper and your valuable suggestions, specifically about the mistake on the power consumption calculations.

References

1. Dalgaty, T. *et al.* Hybrid CMOS-RRAM neurons with intrinsic plasticity. In *International Symposium on Circuits and Systems (ISCAS), 2019* (IEEE, 2019).

REVIEWERS' COMMENTS

Reviewer #3 (Remarks to the Author):

Thank you for the corrections.